# The Development of a Multivalent Capripoxvirus-Vectored Vaccine Candidate to Protect against Sheeppox, Goatpox, Peste des Petits Ruminants, and Rift Valley Fever

**DOI:** 10.3390/vaccines12070805

**Published:** 2024-07-20

**Authors:** Hani Boshra, Graham A. D. Blyth, Thang Truong, Andrea Kroeker, Pravesh Kara, Arshad Mather, David Wallace, Shawn Babiuk

**Affiliations:** 1National Centre for Foreign Animal Disease, Canadian Food Inspection Agency, Winnipeg, MB R3E 3M4, Canada; hboshra@uliege.be (H.B.); thang.truong@phac-aspc.gc.ca (T.T.); andrea.kroeker@usask.ca (A.K.); 2Department of Pathology, Fundamental and Applied Research for Animals and Health (FARAH), Faculty of Veterinary Medicine, University of Liège, 4000 Liège, Belgium; 3ARC-Onderstepoort Veterinary Research, Onderstepoort, Pretoria 0110, South Africa; karap@arc.agric.za (P.K.); mathera@arc.agric.za (A.M.);; 4Department of Immunology, Max Rady College of Medicine, University of Manitoba, Winnipeg, MB R3E 0T5, Canada

**Keywords:** sheeppox, goatpox, lumpy skin disease, peste des petits ruminants, Rift Valley fever, vector, vaccine

## Abstract

Capripoxviruses are the causative agents of sheeppox, goatpox, and lumpy skin disease (LSD) in cattle, which cause economic losses to the livestock industry in Africa and Asia. Capripoxviruses are currently controlled using several live attenuated vaccines. It was previously demonstrated that a lumpy skin disease virus (LSDV) field isolate from Warmbaths (WB) South Africa, ORF 005 (IL-10) gene-deleted virus (LSDV WB005KO), was able to protect sheep and goats against sheeppox and goatpox. Subsequently, genes encoding the protective antigens for peste des petits ruminants (PPR) and Rift Valley fever (RVF) viruses have been inserted in the LSDV WB005KO construct in three different antigen forms (native, secreted, and fusion). These three multivalent vaccine candidates were evaluated for protection against PPR using a single immunization of 10^4^ TCID_50_ in sheep. The vaccine candidates with the native and secreted antigens protected sheep against PPR clinical disease and decreased viral shedding, as detected using real-time RT-PCR in oral and nasal swabs. An anamnestic antibody response, measured using PPR virus-neutralizing antibody response production, was observed in sheep following infection. The vaccine candidates with the antigens expressed in their native form were evaluated for protection against RVF using a single immunization with doses of 10^4^ or 10^5^ TCID_50_ in sheep and goats. Following RVF virus infection, sheep and goats were protected against clinical disease and no viremia was detected in serum compared to control animals, where viremia was detected one day following infection. Sheep and goats developed RVFV-neutralizing antibodies prior to infection, and the antibody responses increased following infection. These results demonstrate that an LSD virus-vectored vaccine candidate can be used in sheep and goats to protect against multiple viral infections.

## 1. Introduction

Infectious diseases of livestock represent a significant threat to animal health, world food supply, and the livelihood of livestock farmers. Small ruminant livestock, such as sheep and goats, are susceptible to devastating viral pathogens with overlapping geographical ranges, including sheeppox virus (SPPV), goatpox virus (GTPV), peste des petits ruminants virus (PPRV), and Rift Valley fever virus (RVFV) [1]. RVFV is of particular concern due to its vector-borne transmission route and zoonotic nature, resulting in human outbreaks with high rates of fatalities [2,3].

SPPV, GTPV, and lumpy skin disease virus (LSDV) are part of the family *Poxviridae* genus capripoxvirus that infect sheep, goats, and cattle, respectively [4,5,6]. SPPV and GTPV preferentially infect sheep and goats, although some strains can infect both species of animals [7]. Morbidity and mortality rates are variable, depending on the animal breed and virus strain, but in severe cases approach 90–100% [8]. Outside of direct animal health concerns, SPPV and GTPV infections decrease animal production yields, decrease milk production, and damage animal skin used for leather production through lesion development that covers most of the animal [9,10]. Geographically, SPPV and GTPV are endemic in Southeast and Central Asia, as well as Northern and Central Africa [11]. Capripoxvirus infections are notifiable diseases according to the World Health Organization for Animal Health (OIE) guidelines.

PPRV is a *Morbillivirus* related to the eradicated Rinderpest morbillivirus. PPRV infects small ruminants, with a mortality rate of up to 70–80% [12]. Infection results in a broad spectrum of symptoms, including fever, ocular and nasal discharge, sores, coughing, diarrhea, and abortions [13,14,15,16]. Additionally, infection with PPRV results in immunosuppression, increasing the risk of infection with other pathogens [17]. PPRV is currently endemic in the Middle East, large regions of Asia, and Central and Northern Africa [18].

RVFV is a vector-transmitted virus belonging to the class *Bunyaviricetes*. RVFV is endemic in Southern and Eastern Africa [19], with outbreaks also occurring in Saudi Arabia, Yemen [20], and Sudan [21]. A wide range of agriculturally important animals may be infected with RVFV, including sheep, goats, and cattle. Symptoms of infection include fever, diarrhea, and high rates of abortion [22]. Additionally, RVFV is a zoonotic pathogen and can cause sporadic human outbreaks, causing significant morbidity and mortality. Over the past 30 years, RVFV has been responsible for hundreds of human deaths and hundreds of thousands of animal deaths [2]. Collectively, SPPV/GTPV, PPRV, and RVFV represent global threats to both human and animal health and the economic well-being of small ruminant farmers [23,24]. Current vaccination strategies are not effective, and novel solutions must be developed to prevent economic losses, disrupt disease transmission resulting in human and animal disease outbreaks, and prevent pathogen spread to disease-free nations [25].

As RVFV, PPRV, and SPPV/GTPV demonstrate a strong degree of geographic co-localization, a multivalent vaccine which would provide protection against all three pathogens would offer both an economic and practical advantage over the current vaccine strategies used in Africa and the Middle East; currently, no such vaccines are in existence [1].

Vaccination strategies for SPPV/GTPV [26,27] and PPRV [28] have previously used live, attenuated viruses, with varied effectiveness in disease prevention. Recently, multiple different RVFV vaccines have been developed for use in animals using a wide range of vaccination strategies [29,30]. The manipulation of poxvirus genomes has previously been used to generate attenuated viruses through the knockout of virulence-associated genes [31] and to generate viral expression vectors that express the protective antigens for RVFV [32,33], PPRV [23,34,35], and foot-and-mouth disease virus [36]. Capripoxviruses represent potential vaccine vectors for vaccination against ruminant pathogens [37,38]. An attenuated LSDV backbone vaccine candidate for use in sheep and goats was generated by knocking out ORF 005 (IL-10 gene) from the virulent LSDV Warmbaths (WB) isolate [39]. This attenuated virus protects sheep and goats against infection with the virulent capripoxvirus.

Here, further development of this capripox vaccine candidate to generate a novel, recombinant capripoxvirus-vectored vaccine expressing the PPRV-F protein and RVFV-GnGc glycoproteins is described. This recombinant vaccine candidate protected sheep against PPRV infection and goats/sheep against RVFV infection. This work demonstrates the efficacy of a single-dose vaccine candidate to protect against three important ruminant pathogens and advances the utility of capripoxvirus-based vaccine vectors for other agriculturally important pathogens [1].

## 2. Materials and Methods

### 2.1. Cells and Viruses

OA3.Ts cells (ATCC CRL-6546) were cultured in Dulbecco’s Modified Eagle’s Medium (DMEM) supplemented with 4 mM L-glutamine, 10% fetal bovine serum (FBS) (Multicell Media-Gibco-BRL-USA), and 1% Penicillin/Streptomycin solution (Multicell). Vero and Vero E6 cells were cultured in DMEM supplemented with 10% FBS. All cells were maintained at 37 °C and 5% CO_2_. The virulent LSDV Warmbaths isolate was used as the parental strain for vaccine generation. LSDV WB and all generated vaccine strains were propagated and titrated on OA3.Ts, as previously described [40].

PPRV Malig (Yemen), originally obtained from The Pirbright Institute (Pirbright, UK), was passaged four times in Vero cells (ATCC CCL-81, Manassas, VA, USA), as previously described [41].

RVFV (Kenyan 2006) was propagated and titrated on Vero E6 cells (ATCC CRL-1586). All work using LSDV, LSDV-vectored prototype vaccines, and PPRV was performed in the CL3 Ag laboratory, and work using RVFV was performed in the CL3 zoonotic laboratory.

### 2.2. Generation of Recombinant Vaccine Candidates

LnRP, LtpaRP, and LfusRP vaccine candidates were generated using previously described techniques [42]. Briefly, plasmid constructs containing PPRVF protein, RVFV-GnGc, eGFP, and gpt sequences flanked by LSDV IL-10 sequences from ORF 005 were generated by gene synthesis (Genscript, Piscataway, NJ, USA) and inserted into the pTW005 vector, as previously described [39]. A schematic representation of the inserts used in LnRP, LtpaRP, and LfusRP is shown in Figure 1. OA3.T cells were seeded in 6-well plates and infected with LSDV WB at an MOI of 0.1. Cells were then transfected with 1.5 μg of each plasmid using X-tremeGENE HP DNA transfection reagent (Roche, Basel, Switzerland) to allow for homologous recombination. After 7 days, the virus was collected and passaged two additional times on OA3.Ts cells in gpt selection media (Sigma, St. Louis, MO, USA). The GFP-positive virus was plaque-purified three times, and the presence of PPRV and RVFV genes and the absence of the LSDV IL-10 gene were screened by PCR. Recombinant viruses were subsequently passaged three times in the absence of gpt selection media to remove the eGFP and gpt selection markers. GFP-negative recombinant viruses were plaque-purified an additional three times and confirmed by PCR.

### 2.3. Immunization and Infection of Animals

Boer cross goats and Rideau Arcott sheep were used in the animal experiments, which were conducted under the approval of the Canadian Science Centre for Human and Animal Health Animal Care Committee, which follows the guidelines of the Canadian Council on Animal Care.

Sheep and goats were acclimated to CL3 animal cubicles at the National Centre for Foreign Animal Disease (NCFAD, Winnipeg, Canada) and monitored daily for signs of disease. The scoring of clinical signs was performed by the attending technicians and veterinarians with no prior knowledge of which animals belonged to which experimental group, thereby conducting their observations blind to each vaccination group.

For PPRV studies, after acclimatization, sheep were vaccinated by intradermal inoculation of 0.1 mL containing 10^4^ TCID50 of either LnRP, LtpaRP, or LfusRP vaccine candidates in DMEM.

Sheep were challenged with Malig PPRV using 2 mL delivered intranasally and 2 mL through subcutaneous injection, using a virus stock titrated at 10^4.5^ TCID50/mL, as previously described [41]. All animals were observed daily, with clinical signs recorded throughout the study. Rectal temperatures were measured daily from 1 to 13 days post-infection (dpi). Oral and nasal swabs, as well as blood, were collected from sheep and goats 2 days prior to infection, and at 2, 4, 6, 8, 11, 13, 15, 18, and 21 dpi.

For RVFV studies, sheep and goats were immunized by intradermal inoculation of 0.1 mL containing either 10^4^ or 10^5^ TCID50 of the LnRP vaccine candidate. Sheep and goats were infected with RVFV by subcutaneous inoculation with 1 mL of RVFV at a dose of 10^6^ TCID50 (Kenya, 2006) 24 days following immunization. Sera samples were collected at 1, 2, 3, 4, 5, 7, 9, 11, 14, and 21 dpi to measure viremia and neutralizing antibody production.

### 2.4. Virus Neutralization Assays

Sera samples were collected from animals at the indicated time points and serially diluted with DMEM. Diluted sera samples were then incubated with 100 TCID50 of PPRV or RVFV and incubated at 37 °C for 1 h. Virus and sera samples were then added to confluent Vero E6 monolayers in 96-well plates for PPRV or in 24-well plates for RVFV and incubated at 37 °C for 1 h. Cells were then washed and incubated in fresh media (TCID) or overlay media (plaque assay). Viral replication was assessed by monitoring the cytopathic effect (TCID) for PPRV and by plaque assay for RVFV [43]. Neutralizing antibody titres for PPRV were expressed as the inverse dilution where virus neutralization occurred, and for RVFV, they were expressed as the inverse dilution where a 70% reduction in plaque numbers occurred.

### 2.5. Viral Shedding

Viral copy number was determined from oral and nasal swabs and blood samples using reverse transcriptase quantitative PCR (RT-qPCR). RNA was isolated from the samples using the QIAamp Viral RNA Mini Kit (Qiagen, Hilden, Germany), and RT-qPCR was performed using previously described protocols for PPRV [41] and RVFV [44]. RVFV viremia was detected by incubating sera samples on confluent Vero E6 layers to propagate a live RVFV.

### 2.6. Controls

For infection experiments, negative control animals were inoculated with DMEM. For viral genome quantification, viremia determination, and viral load experiments, tissues from uninfected animals were used as the negative control.

## 3. Results

### 3.1. Generation of Three Multivalent Capripox-Vectored Vaccine Candidates Expressing Different Antigen Forms of PPRV F and RVFV GnGc

Previous studies have determined that deletion of the IL-10 locus from the capripoxvirus genome can attenuate the virus and generate a live attenuated capripoxvirus vaccine in sheep and goats [39]. Three recombinant capripoxvirus-vectored vaccine candidates were generated using homologous recombination to delete the IL-10 locus, while simultaneously inserting either (1) the full-length PPRV-F protein and RVFV-GnGc glycoprotein (LnRP); (2) the full-length PPRV-F protein and RVFV-GnGc glycoprotein with a secretory signal sequence on the N-terminus (LtpaRP); or (3) a single fusion protein containing domains from both the PPRV-F protein and RVFV-GnGc glycoprotein (LfusRP) (Figure 1). Recombinant viruses were generated in vitro, and the presence of PPR and RVFV gene expression was confirmed by RT-qPCR and immunocytochemistry.

### 3.2. Recombinant Multivalent Capripox-Vectored Vaccine Candidates Protect against PPR in Sheep

To assess if the LnRP, LtpaRP, or LfusRP vaccine candidates prevent infection in sheep with PPRV, sheep (n = 6) were vaccinated with either DMEM (control) or 10^4^ TCID50 of the LnRP, LtpaRP, or LfusRP vaccine candidates and infected 21 days post-immunization with PPRV. Vaccinated animals did not develop clinical signs of capripoxvirus infection and developed only slight, sporadic fever after immunization (Figure 2a).

Unvaccinated sheep developed overt clinical signs of PPRV infection, including malaise and oral/nasal discharge, and two unvaccinated animals were euthanized due to clinical disease reaching the endpoint of the study. In contrast, none of the animals vaccinated with LnRP or LtpaRP developed clinical signs of disease and had a 100% survival rate. Control and LfusRP-vaccinated sheep developed fever 4 days post-infection, with fever persisting for 13 days post-infection (Figure 2a). In contrast, LnRP- and LtpaRP-vaccinated animals did not develop fever after infection with PPR (Figure 2a). While vaccination alone with any of the three vaccines did not induce detectable levels of PPRV-neutralizing antibodies (Figure 2b), all animals developed neutralizing antibodies following the PPRV challenge. While LfusRP-vaccinated animals developed neutralizing antibodies 2 days earlier than control animals, these animals displayed persistent fever (Figure 2b).

Control animals had significant viral genome shedding in oral secretions from 4 to 10 dpi (Figure 3a). PPRV was only detected in LnRP- and LtpaRP-vaccinated animals at 6 days post-infection in one and two animals, respectively (Figure 3b,c). LfusRP-vaccinated sheep displayed PPRV shedding in oral secretions like control sheep (Figure 3d). Nasal secretions displayed a similar pattern, with control vaccinated animals showing significant viral shedding from 4 to 10 dpi (Figure 3e), whereas PPRV shedding in oral secretions in LnRP- and LtpaRP-vaccinated animals was only detected on day 6 post-infection (Figure 2f,g). LfusRP-vaccinated animals displayed PPRV shedding in oral secretions from 4 to 10 dpi, like control vaccinated animals (Figure 3h). Collectively, these results demonstrate the effectiveness and safety of two capripox-vectored vaccine candidates (LnRP and LtpaRP) and the failure of LfusRP against PPRV infection in sheep.

### 3.3. A Recombinant Multivalent Capripox-Vectored Vaccine Candidate Protects against RVFV in Sheep and Goats

To further explore the ability of a multivalent vaccine candidate to protect against RVFV, sheep and goats were vaccinated with two different doses of the vaccine candidate containing both native forms of the PPRV-F protein and RVFV-GnGn glycoprotein (LnRP) since there was no difference between the native and secreted forms of the RVFV-GnGn glycoprotein with PPRV infection. Sheep and goats were immunized with DMEM (control) or with either 10^4^ TCID_50_ or 10^5^ TCID_50_ of the LnRP vaccine candidate and subsequently infected with RVFV 24 days post-immunization. After RVFV infection, control animals developed sporadic fever, whereas no fever was detected in immunized animals (Figure 4). This sporadic fever in infected sheep and goats is consistent with rectal temperatures using an established RVFV infection model in sheep and goats [45]. Sheep and goats vaccinated with both a low and high dose of LnRP developed RVFV-neutralizing antibodies prior to RVFV infection, beginning at 7 days post-immunization. After RVFV infection, sheep vaccinated with both a low and high dose of LnRP developed higher neutralizing antibodies than unimmunized sheep. Goats immunized with either the low or high dose developed higher neutralizing antibody titres on days 9 and 11 post-infection. Despite the qualitative data represented in Figure 4, it should be noted that no statistical significance was observed, likely due to the small sample size in each experimental group.

RVFV was detected in sera by virus isolation in all unvaccinated sheep on day 1 post-infection (Figure 4a) and in five out of six unvaccinated goats on day 1 post-infection (Figure 5b). In contrast, sheep and goats vaccinated with both the low dose and the high dose of LnRP had undetectable viremia. However, while infectious RVFV was undetectable from the vaccinated groups, the quantitative RT-qPCR results showed their viral copy numbers to be comparable to those of the control group in both sheep (Figure 5c) and goats (Figure 5d), suggesting that viral neutralization occurred within the first 4 days of viral infection. Taken together, these results show the ability of a single-dose capripoxvirus-vectored vaccine candidate (LnRP) to protect sheep and goats against RVFV and prevent viable viruses from being isolated in sera.

## 4. Discussion

These results demonstrate the successful generation of a multivalent vaccine candidate to protect sheep and goats against SPPV/GTPV, PPRV, and RVFV; protective immunity occurred after a single dose of a live, attenuated capripox-vectored vaccine candidate. Recently, the LSDV WB005KO vaccine construct was evaluated in cattle, with its protective efficacy being evaluated against LSDV infection [46]. While the vaccine candidate was protective, the LSDV WB005KO vaccine backbone was found not to be sufficiently attenuated, as evidenced by the adverse reactions observed in the vaccinated animals. However, using this same construct in sheep and goats, no severe secondary reactions were observed. Further evaluation of LSDV WB005KO is required to ensure that the vaccine is safe and does not spread from sheep or goats into cattle. This is important since with LSDV, there have been reports of contaminated vaccines generating virulent recombinant LSDV currently spreading in Asia [47,48]. Additionally, not fully attenuated Kenyan sheep- and goatpox vaccines are currently causing LSDV in the Indian subcontinent [49,50].

Three potential vaccine candidates were evaluated (LnRP, LtpaRP, and LfusRP), containing different variations of the PPR-F protein and RVFV-GnGc glycoprotein, for their ability to prevent PPRV infections in sheep. Immunization alone with all three of the vaccine candidates did not generate PPRV-neutralizing antibodies following immunization. However, following PPRV infection, vaccinated animals all generated neutralizing antibodies earlier in comparison to unvaccinated animals. Both the LnRP and LtpaRP vaccine candidates prevented fever and PPRV shedding, whereas the LfusRP-vaccinated animals presented with fever and significant viral shedding after PPRV infection. This indicates that the recombinant fusion protein was not able to generate a protective immune response in vaccinated sheep. Following immunization with LnRP or LtpaRP, it is hypothesized that immune mechanisms such as T-cell activation or non-neutralizing antibody activation (via the complement system or interaction with Fc receptors) provide an initial priming response, which, in concert with the neutralizing antibodies generated following infection with PPRV, provide a more robust immune response. The expression of the full-length PPRV-F protein and RVFV-GnGc glycoproteins in our vaccine candidates protected sheep from PPRV infection, regardless of whether the RVFV-GnGc glycoproteins expressed proteins containing a secretory signal or not. Taken together, these results indicate that the expression of the PPRV-F protein in attenuated capripoxviruses generates protective immune responses against PPRV, and the expression of additional viral antigens does not interfere with the protective immune response.

The LnRP vaccine candidate, expressing the native forms of the PPRV-F protein and RVFV-GnGc glycoprotein, prevented RVFV infection in both sheep and goats. The vaccine candidate was able to elicit neutralizing RVFV antibody responses and prevent RVFV viremia. Collectively, these results show a single immunization with the attenuated capripoxvirus-vectored LnRP vaccine candidate protects against PPRV and RVFV infection.

The design of this LnRP vaccine candidate offers several advantages. First, it is the first multivalent vaccine candidate demonstrated to provide protection against SPPV/GTPV, PPRV, and RVFV infection. Second, serological diagnostics for PPRV and RVFV do not use the PPRV F protein and RVFV-GnGc antigens present in the vaccine candidate. This LnRP vaccine candidate has the ability to differentiate infected from vaccinated animals (DIVA) for use in PPRV and RVFV disease-free nations. Third, CaPV genomes can be easily modified to allow for the future generation of novel vaccines using CaPV as a vector.

## 5. Conclusions

PPRV alone threatens approximately 80% of the world’s population of sheep and goats, causing USD 1.45-2.1 billion in economic losses per year [51]. Increased legal and illegal trade of animals and climate change have all recently contributed to the expanding range of SPPV/GTPV, PPRV, and RVFV. Previous outbreaks of foot-and-mouth disease (FMD), bluetongue (BT), the highly pathogenic avian influenza (HPAI), and Newcastle disease have demonstrated the economic impact of viral epidemics on the livestock and poultry industries [52,53]. Therefore, it is important to have multivalent vaccines (which allow for the distinction between vaccinated and infected animals) available [54]. For small ruminants, access to vaccines for SPPV/GTPV, PPRV, and RVFV is critical for many regions in Africa. This study aimed to develop novel approaches to control multiple important animal diseases in a single vaccine. Further development and testing of new approaches to control or eradicate SPPV/GTPV, PPRV, and RVFV are essential in addressing the increasing threat and risk that these pathogens pose to both human and animal health [1]. The data provided in this study demonstrate that a multivalent vaccine candidate that can protect against all three pathogens (SPPV/GTPV, PPRV, and RVFV) is feasible.

Previous work using this vaccine backbone (i.e., LSDV WB005KO) demonstrated that this gene-deleted virus was able to protect sheep and goats against sheeppox and goatpox [39]. However, due to ethical and practical limitations, this study was not able to evaluate the degree of mutual immune interference caused by the infection of a virulent strain of one pathogen followed by another infection from another virulent strain of another pathogen (i.e., capripox infection, followed by PPRV infection, followed by RVFV infection). Field studies in areas where all three pathogens are endemic will provide insights into the efficacy of sequential infection, as well as determine if mutual interference occurs. In addition, further work is required to determine the duration of immunity and for licensing of the candidate vaccine.

## Figures and Tables

**Figure 1 vaccines-12-00805-f001:**
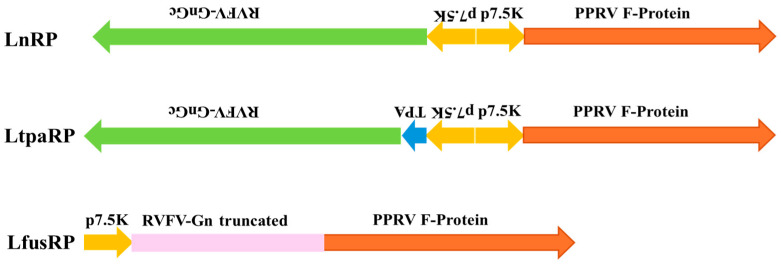
A schematic illustration of the 3 vaccine candidates used in this study. In all 3 contructs, the cDNA encoding for the fusion protein (F protein) of PPRV was expressed under the vaccinia 7.5K promoter. In the case of LnRP, the cDNA encoding for the GnGc glycoprotein precursor was expressed in the opposite orientation, also using the vaccinia 7.5K promoter. For LtpaRP, a construct similar to LnRP was used, differing only in the addition of a TPA signal sequence upstream of the RVFV GnGc ORF. For LfusRP, a fusion protein was generated using the soluble portion of RVFV Gn, immediately followed by the full-length PPRV F-protein sequence.

**Figure 2 vaccines-12-00805-f002:**
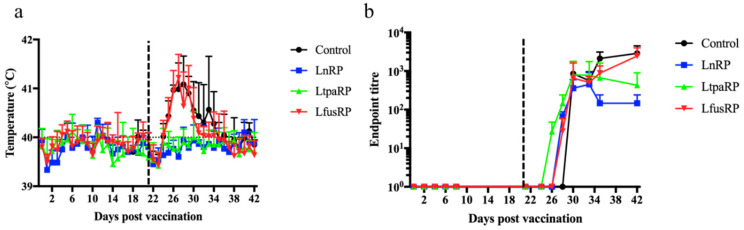
Capripox–PPR-RVFV multivalent vaccine candidates protect sheep against PPR infection. Sheep were vaccinated with either LnRP, LtapRP, or LfusRP vaccine candidates and infected with PPR at 21 days post-immunization, indicated by the dashed line. Rectal temperatures (**a**) and neutralizing antibodies against PPRV (**b**) were measured on each indicated day. Mean values are displayed with the standard deviation.

**Figure 3 vaccines-12-00805-f003:**
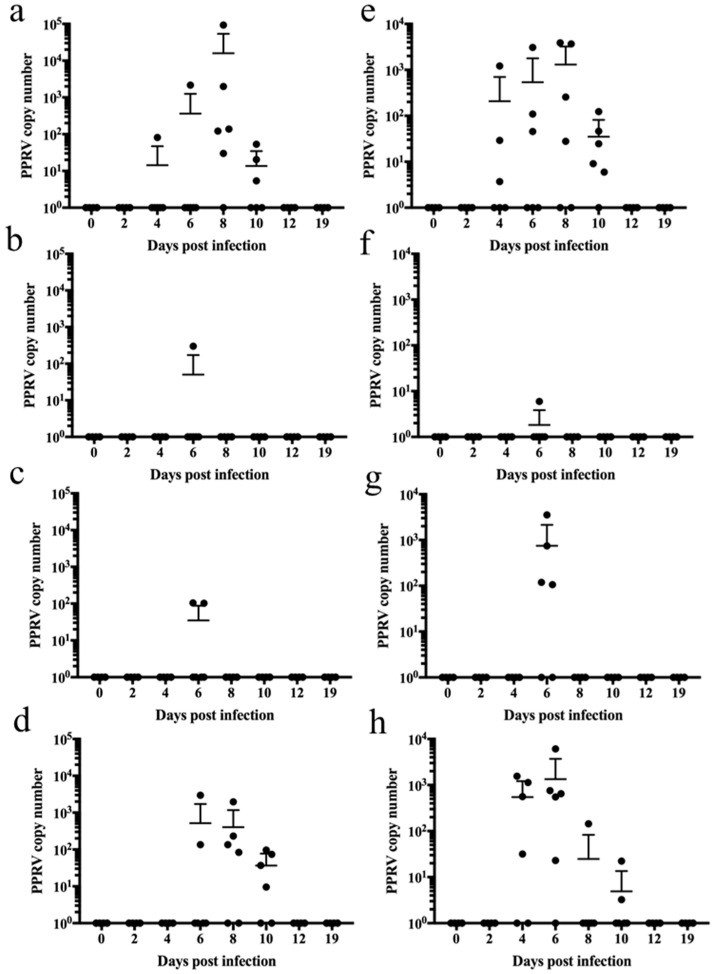
Capripox–PPR-RVFV multivalent vaccine candidates LnRP and LtapRP prevent shedding of PPRV in oral and nasal secretions in sheep. PPRV genome copies in oral swabs from (**a**) unvaccinated controls and (**b**) LnPR-vaccinated, (**c**) LtapRP-vaccinated, and (**d**) LfusRp-vaccinated animals. PPRV genome copies in nasal swabs from (**e**) unvaccinated controls and (**f**) LnPR-vaccinated, (**g**) LtapRP-vaccinated, and (**h**) LfusRp-vaccinated animals. RNA was extracted, and the PPRV copy number was determined by RT-qPCR.

**Figure 4 vaccines-12-00805-f004:**
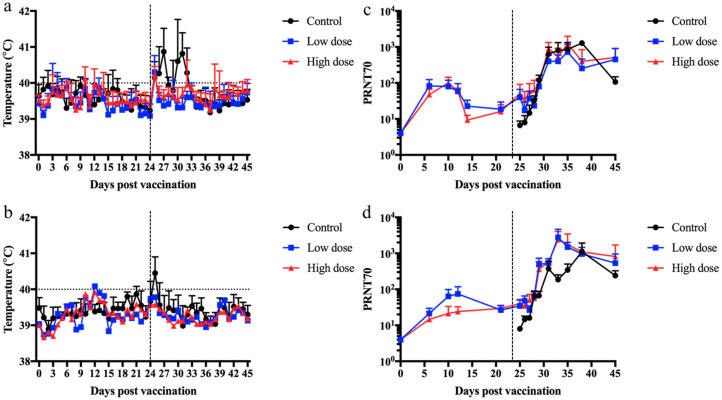
The capripox–PPR-RVFV multivalent vaccine candidate LnRP protects sheep and goats against RVFV infection and elicits neutralizing antibodies. Sheep and goats were vaccinated with a low dose or a high dose of the LnRP vaccine candidate and subsequently infected with RVFV at 24 days post-immunization. Rectal temperatures of sheep (**a**) and goats (**b**) were taken on each indicated day. Sera samples were collected on each indicated day, and RVFV-neutralizing antibodies were detected in sheep (**c**) and goat (**d**) samples. Mean values are displayed with the standard deviation.

**Figure 5 vaccines-12-00805-f005:**
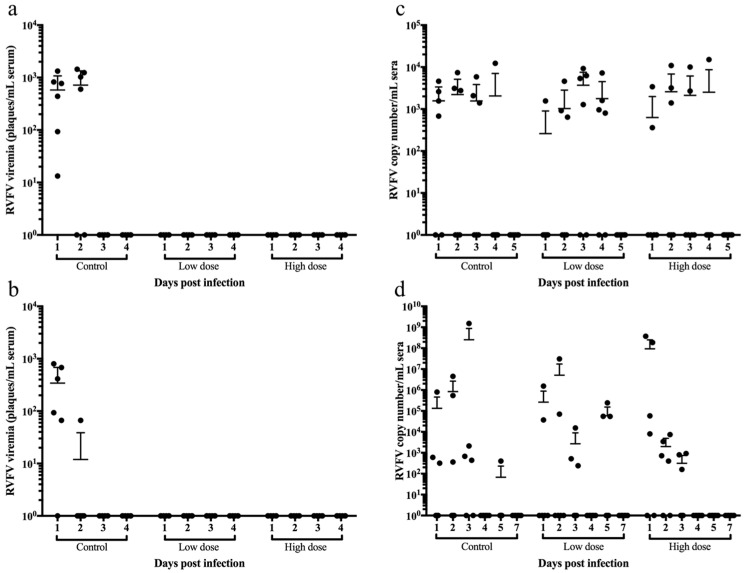
The capripox–PPR-RVFV multivalent vaccine candidate LnRP protects sheep and goats against RVFV viremia. Sheep were vaccinated with a low dose or a high dose of the LnRP vaccine candidate and challenged with RVFV at 24 days post-immunization. The virus was quantified from infected sheep (**a**) and goats (**b**) at each time point indicated. Sera were collected from control, low dose, and high dose vaccinated sheep (**c**) and goats (**d**), and the RVFV copy number was determined by RT-qPCR on each day post-infection.

## Data Availability

The datasets used and/or analyzed during the current study are available from the corresponding author on reasonable request.

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
