# Peer review of "The Development of a Multivalent Capripoxvirus-Vectored Vaccine Candidate to Protect against Sheeppox, Goatpox, Peste des Petits Ruminants, and Rift Valley Fever"

_vaccines, 2024, doi:10.3390/vaccines12070805_

Round 1

Reviewer 1 Report

Comments and Suggestions for Authors

The submitted manuscript, Development of a multivalent capripoxvirus vectored vaccine 2 to protect against sheeppox, goatpox, peste des petits ruminants 3 and Rift Valley fever, proposes new constructs for vaccines to protect livestock from diseases that currently have no suitable vaccine available. The data is promising and suggests these constructs are worth pursuing to vaccine production. The approach was enhanced by including three separate formulations of antigen expression and by using antigens not used in diagnostic testing.

There are a few weaknesses in how the data is presented/interpreted:

1) In Section 3.2, lines 195 and 196 suggest that the presence of neutralizing antibodies 2 and 4 days earlier than control animals is important. However, this was true for the LfusRP construct as well (Figure 1b), and the authors have acknowledged that this construct is not successful. Additionally, it was the negative control group that reached the highest titer on the last reported timepoint (while the most successful constructs dropped off earlier), which calls into question how much value can be placed on the timing of the antibodies. It does not relate particularly well to Figure 1a results show that the LfusRP construct, though producing antibodies earlier than the control group was quite similar to the control group in elevated temperatures.

2) On line 205, the text states, “Control animals had significant viral shedding in oral secretions from day 4 to day 10 dpi (Figure2a).” The measurement, however, is genome copies, not detection of infectious virus. In Figure 4, the results demonstrate that genome copies does not correlate to viremia. Please clarify if the viral shedding was determined to be infectious virus particles.

3) On lines 230-232, the text states that there was no fever detected in vaccinated animals, while controls developed sporadic fever. While it is obvious that there is a difference shown in the figure between the controls and vaccinated animals, it is not strictly true that there was no fever at all with the two vaccinated groups, and it is concerning that the animals receiving the higher dose had more frequent slight fever elevations than those receiving the lower dose. And again, adding p values would help to clarify significance.

In general, the figures were presented clearly. However, Figure 2 is too small to read without zooming in.

The statement on lines 71-74 regarding current vaccine strategies should have a relevant citation(s) included.

Please check the reference cited at the bottom of page 2 (line 96). It may not be the one you intended to include at that point.

I believe that your interpretation of reference number 34 is far beyond what the authors of that paper have expressed. Please reword the lines 309 to 311 of your discussion to more accurately reflect the citation’s message or find a different citation to support your conclusion.

Comments on the Quality of English Language

The English has only minor errors, some of which could even be typos.

Author Response

The submitted manuscript, Development of a multivalent capripoxvirus vectored vaccine to protect against sheeppox, goatpox, peste des petits ruminants and Rift Valley fever, proposes new constructs for vaccines to protect livestock from diseases that currently have no suitable vaccine available. The data is promising and suggests these constructs are worth pursuing to vaccine production. The approach was enhanced by including three separate formulations of antigen expression and by using antigens not used in diagnostic testing.

There are a few weaknesses in how the data is presented/interpreted:

The Authors would like to thank Reviewer #1 for his/her comments.  We hope that we have sufficiently addressed his/her concerns to his/her satisfaction.

1) In Section 3.2, lines 195 and 196 suggest that the presence of neutralizing antibodies 2 and 4 days earlier than control animals is important. However, this was true for the LfusRP construct as well (Figure 1b), and the authors have acknowledged that this construct is not successful. Additionally, it was the negative control group that reached the highest titer on the last reported timepoint (while the most successful constructs dropped off earlier), which calls into question how much value can be placed on the timing of the antibodies. It does not relate particularly well to Figure 1a results show that the LfusRP construct, though producing antibodies earlier than the control group was quite similar to the control group in elevated temperatures.

The Authors agree with the Reviewer with regards to the inconclusive results of LfusRP.  As the Reviewer pointed out, while the LfusRP-vaccinated animals did induce antibodies 2 days prior to the control animals, LfusRP-vaccinated animals still exhibited elevated temperatures.    However, the Authors should point out that the general antibody profile of the LfusRP group was similar to that of the control group.  If anything, this should confirm that the protective efficacy of the LfusRP group is similar to the negative control (i.e. no immune protection was generated prior to virus challenge).  The fact that both LnRP and LtpaRP protected the animals from fever, as well as providing less neutralizing antibodies suggests that the fusion protein induces immunity through other mechanisms apart from neutralizing Abs.  The authors propose that following vaccination with LnRP or LtpaRP, immune mechanisms such as T-cell activation or non-neutralizing antibody activation (via the complement system or interaction with Fc receptors) provide an initial priming response, which in concert with the neutralizing antibodies generated following challenge with PPRV, provided a more robust immune response.  In order to address the Reviewer’s concern, the Authors have added the following to line 292 of the discussion section,

“ We hypothesize that following vaccination with LnRP or LtpaRP, immune mechanisms such as T-cell activation or non-neutralizing antibody activation (via the complement system or interaction with Fc receptors) provide an initial priming response, which in concert with the neutralizing antibodies generated following challenge with PPRV, provided a more robust immune response.”

2) On line 205, the text states, “Control animals had significant viral shedding in oral secretions from day 4 to day 10 dpi (Figure2a).” The measurement, however, is genome copies, not detection of infectious virus. In Figure 4, the results demonstrate that genome copies does not correlate to viremia. Please clarify if the viral shedding was determined to be infectious virus particles.

While the Authors agree with the Reviewer that measuring infectious particles is usually a better method to detect infectious virus particles such as RVFV, we were unable to measure PPRV particles in unvaccinated animals following PPRV challenge at any timepoint (despite observing clinical signs), thereby suggesting that virus titration of experimental PPRV was more difficult than RVFV.  This might be due to limitations or difficulty in isolation of PPRV using diluted mucosal swabs.  Nevertheless, the authors hope that the genomic data can still provide strong evidence for protection in immunized animals.       

3) On lines 230-232, the text states that there was no fever detected in vaccinated animals, while controls developed sporadic fever. While it is obvious that there is a difference shown in the figure between the controls and vaccinated animals, it is not strictly true that there was no fever at all with the two vaccinated groups, and it is concerning that the animals receiving the higher dose had more frequent slight fever elevations than those receiving the lower dose. And again, adding p values would help to clarify significance.

While the Authors agree that p-values would help clarify significance, the number of animals in each group, along with the natural variation in body temperatures resulted in p > 0.05.  Therefore, we were only able to qualitatively point out the general trend of the vaccinated group showing less fever than the control group.  While the Authors also agree with the Reviewer with regards to the higher dose group having slightly more fever than the lower dose group, we were (once again) unable to statistically demonstrate that.

In general, the figures were presented clearly. However, Figure 2 is too small to read without zooming in.

The Authors agree, and have made Figure 2 significantly bigger in order to make it easier to read.

The statement on lines 71-74 regarding current vaccine strategies should have a relevant citation(s) included.

The Authors agree, and have included citation [15] in the sentence, as it extensively covers the strategies used to combat PPRV, RVFV and capripoxviruses.

Please check the reference cited at the bottom of page 2 (line 96). It may not be the one you intended to include at that point.

The Reviewer is correct.  The Authors meant to use reference 15, not 25.

I believe that your interpretation of reference number 34 is far beyond what the authors of that paper have expressed. Please reword the lines 309 to 311 of your discussion to more accurately reflect the citation’s message or find a different citation to support your conclusion.

The Authors agree.  In order to better reflect reference 34, the Authors have completely changed the sentence.  It is now the following:

“Previous outbreaks of foot-and-mouth disease (FMD), bluetongue (BT), the highly pathogenic avian influenza (HPAI) and Newcastle disease have demonstrated the economic impact of viral epidemics on the livestock industry.  Therefore, the importance of having multivalent vaccines (which allow the distinction between vaccinated and infected animals) should be made easily available, with instant access to vaccines such as peste des petits ruminants and Rift valley fever, among others [34]”. 

Reviewer 2 Report

Comments and Suggestions for Authors

Authors conducted animal experiments to evaluate whether a single dose of vectored live vaccine (backboned by a LSD IL-10 gene deleted virus) carrying protein of PPRV and RVFV are protective of the three diseases.

The title of this manuscript is quite encompassing.  In other words it may be perceived that  a single dose of multivalent vaccine can protect all three diseases evaluated at the same time.  But in fact, the protection against sheeppox and goatpox was based on a previous published paper (If I understand correct). In addition, the protection against PPRV and RVF was conducted in two separate experiments.  One concern is that if protection against all three diseases were not evaluated in the same animal experiment, is there any possibility of mutually interference? or can we make an encompassing statement, such as in this manuscript, combining results of three separate experiments.

section 2.2 and section 3.1: To be able to generate a multivalent vectored vaccine and test it on animals is good, not to mention in 3 different antigenic forms.  There are a few interesting technical details, for example, what is the difference between LnRP and LfusRP and characterization of their products.  I would encourage authors present a schematic diagram of these constructs and recombinant viruses and explain a little bit, even if authors may have other plan for these data.

Author Response

Authors conducted animal experiments to evaluate whether a single dose of vectored live vaccine (backboned by a LSD IL-10 gene deleted virus) carrying protein of PPRV and RVFV are protective of the three diseases.

The title of this manuscript is quite encompassing.  In other words it may be perceived that a single dose of multivalent vaccine can protect all three diseases evaluated at the same time. But in fact, the protection against sheeppox and goatpox was based on a previous published paper (If I understand correct). In addition, the protection against PPRV and RVF was conducted in two separate experiments. One concern is that if protection against all three diseases were not evaluated in the same animal experiment, is there any possibility of mutually interference? or can we make an encompassing statement, such as in this manuscript, combining results of three separate experiments.

The Authors would like to thank Reviewer #2 for his/her time in going over the manuscript.  With regards to Reviewer #2’s previous comment, we agree with the statement that, as capripox immunity was evaluated in a previous manuscript, the authors cannot make a definitive statement about mutual interference.  Therefore, the Authors have added an encompassing statement (as suggested).  It has been added to last paragraph in the Conclusion section:

“Previous work using this vaccine backbone (i.e. LSDV WB005KO) demonstrated that this gene deleted virus was able to protect sheep and goats against sheeppox and goatpox [24].  However, due to ethical and practical limitations, we were not able to evaluate the degree of mutual immune interference caused by the challenge of a virulent strain of one pathogen, followed by another challenge from another virulent strain of another pathogen (i.e. capripox challenge, followed by PPRV challenge, followed by RVFV challenge).  Field studies in areas where all three pathogens are endemic will provide some insight into the efficacy of sequential infection, as well as the possibility of mutual interference.” 

section 2.2 and section 3.1: To be able to generate a multivalent vectored vaccine and test it on animals is good, not to mention in 3 different antigenic forms.  There are a few interesting technical details, for example, what is the difference between LnRP and LfusRP and characterization of their products.  I would encourage authors present a schematic diagram of these constructs and recombinant viruses and explain a little bit, even if authors may have other plan for these data.

The Authors agree with Reviewer #2, in that a schematic would help in explaining the difference between the vectors.  Therefore, the Authors have added an addition Supplemental Figure 1 containing a schematic of all three constructs. 

Reviewer 3 Report

Comments and Suggestions for Authors

The authors present an interesting study where they have constructed three LSDV variants that carry protective antigens from PPRV and RVFV that were expressed as either individual proteins or as a fusion protein. The antibody responses of sheep following immunisation and challenge with PPRV were then determined, as well as protection from the expression of clinical signs associated with PPR disease. While antibodies were not detected until, after challenge, immunised animals were protected from clinical disease.

The capacity of the recombinant LSDV viruses to protect sheep and goats from RVF challenge was also evaluated. With the prototype vaccines protecting the animals from expression of clinical disease.

Overall the study is well written, mostly the results are clearly presented and support the conclusions drawn by the authors. 

Throughout the manuscript, the authors use the terms "vaccine", "vaccination" in reference to their LSDV constructs and "infection" when testing the efficacy. I think that it would be more appropriate to use "prototype vaccine" in place of "vaccine", "immunisation" instead of "vaccination" and "challenge" instead of "infection". Each of these terms has specific meanings in the context of this field of research.

Specific comments and suggestions for the authors to consider:

Line 92 Delete repeated words “against PPRV challenge”

Line 136 I would suggest using the term “challenge” rather than “infected” in this context.

Line 161 suggest using the term "reverse transcriptase quantitative PCR (RT-qPCR) here and throughout the manuscript.

Line 168-171 suggest deleting this sentence as it is more suited to the introduction, than the results.

Line 180 The authors should consider including the data regarding transgene expression as supplemental files.

Line 217 Delete repeated “Figure 2.”

Line 183 suggest revision “sheep (n=6)”

Line 187 I would suggest the authors replace “significant” with “overt” or similar wording. The data presented in the manuscript have not been subjected to statistical analyses and as such the term “significant” should not be used.

Were any statistically analyses performed on the data sets? While it can be difficult to tell just looking at a graph, some of the treatments in Fig 1a appear to be quite different. Indeed, a statistical analysis would help avoid this subjective interpretation of the data.

Line 200 Figure 1. The compiling of all temperature data for the treatment groups into one figure makes it very difficult to determine what the data is showing. I would suggest splitting the results onto separate graphs which for each vaccine treatment with the data for the control group on each.

Line 237 Again significance is referred to, but not supported by an appropriate statistical analysis.

Line 241 Figure 3a and 3b – see comments for Fig 2.

Line 278 suggest deletion of “First,”

Line 308 please add a suitable reference(s) to support this statement regarding the expanding range of these viruses.

Line 311-315 Rather than repeating the aims of the study here, I would suggest that the authors write a conclusion that answers the stated aims of the study.

Comments on the Quality of English Language

See comments to authors.

Author Response

The authors present an interesting study where they have constructed three LSDV variants that carry protective antigens from PPRV and RVFV that were expressed as either individual proteins or as a fusion protein. The antibody responses of sheep following immunisation and challenge with PPRV were then determined, as well as protection from the expression of clinical signs associated with PPR disease. While antibodies were not detected until, after challenge, immunised animals were protected from clinical disease.

The capacity of the recombinant LSDV viruses to protect sheep and goats from RVF challenge was also evaluated. With the prototype vaccines protecting the animals from expression of clinical disease.

Overall the study is well written, mostly the results are clearly presented and support the conclusions drawn by the authors.

The Authors would like to thank Reviewer #3 for his positive comments, and have made every effort to address the concerns raised by him/her.  All suggested changes have been made to the manuscript, as indicated in the Track Changes tab. 

Throughout the manuscript, the authors use the terms "vaccine", "vaccination" in reference to their LSDV constructs and "infection" when testing the efficacy. I think that it would be more appropriate to use "prototype vaccine" in place of "vaccine", "immunisation" instead of "vaccination" and "challenge" instead of "infection". Each of these terms has specific meanings in the context of this field of research.

The Authors agree with this suggestion.  As indicated, “vaccine” has been replaced with “vaccine candidate” when talking about the immune agent in question.  “Vaccination” has been replaced with “immunization”, and “challenge” has been changed to “infection” (when discussing the experiments).

Specific comments and suggestions for the authors to consider:

Line 92 Delete repeated words “against PPRV challenge”….Changed

Line 136 I would suggest using the term “challenge” rather than “infected” in this context…..Changed

Line 161 suggest using the term "reverse transcriptase quantitative PCR (RT-qPCR) here and throughout the manuscript…..Changed

Line 168-171 suggest deleting this sentence as it is more suited to the introduction, than the results….Removed

Line 180 The authors should consider including the data regarding transgene expression as supplemental files…The pictures for the transgene expression are not of publication quality so we would prefer not to include them.

Line 217 Delete repeated “Figure 2.”….Removed

Line 183 suggest revision “sheep (n=6)”….Changed

Line 187 I would suggest the authors replace “significant” with “overt” or similar wording. The data presented in the manuscript have not been subjected to statistical analyses and as such the term “significant” should not be used...Changed to “overt”

Were any statistically analyses performed on the data sets? While it can be difficult to tell just looking at a graph, some of the treatments in Fig 1a appear to be quite different. Indeed, a statistical analysis would help avoid this subjective interpretation of the data.  Unfortunately, due to the small sample size, as well as the relatively small change in temperature (a difference of about 1-1.5oC between protected and unprotected animals), we were not able to achieve a p-value of less than 0.05. 

Line 200 Figure 1. The compiling of all temperature data for the treatment groups into one figure makes it very difficult to determine what the data is showing. I would suggest splitting the results onto separate graphs which for each vaccine treatment with the data for the control group on each.

In order to keep the formatting of the paper with the figures, we preferred to keep the figure as is.

Line 237 Again significance is referred to, but not supported by an appropriate statistical analysis….the word “significantly” has been removed

Line 241 Figure 3a and 3b – see comments for Fig 2….The legend from Figure 3 has been verified.

Line 278 suggest deletion of “First,”….Removed

Line 308 please add a suitable reference(s) to support this statement regarding the expanding range of these viruses….The Authors have added reference [15] to this sentence.

Line 311-315 Rather than repeating the aims of the study here, I would suggest that the authors write a conclusion that answers the stated aims of the study...The Authors have added the following sentence, “The data provided in this study demonstrates that a multivalent vaccine candidate that can protect against all three of the aforementioned pathogens is feasible.”

Reviewer 4 Report

Comments and Suggestions for Authors

The objective of this study was to evaluate the efficacy of a novel, recombinant capripoxvirus-vectored vaccine.

1.      Please provide details of the general work frame for this study and also please describe the working hypothesis for the study.

2.      Please provide a detailed description of all controls employed in this study. Please describe control chemicals, control animals, control ‘vaccine’, etc. These must be included in a separate sub-section in M&M.

3.      How did you select and how did you include animals in the study? Please give a list of the criteria.

4.      Visualization for this manuscript is poor. You need to include more graphs with the results and also tables, whilst at the same time you must reduce the text in the manuscript.

5.      Discussion. The Discussion is a bit shallow and does not fully cover all the findings of the study. Please expand and please describe in greater details the clinical benefits from using this vaccine.

6.      Do you think that the vaccine can be commercially available for use by the end of October? This will be very helpful for farmers.

7.      References are OK.

8.      Concluding section is missing.

Overall. Manuscript that can advance to next stage after extensive revision.

Author Response

The objective of this study was to evaluate the efficacy of a novel, recombinant capripoxvirus-vectored vaccine.

The Authors would like to thank Reviewer #4 for his/her time in reviewing this manusript.

  1. Please provide details of the general work frame for this study and also please describe the working hypothesis for the study. The Authors propose that, as previous work using poxviruses have demonstrated that antigens against RVFV and PPRV can be individually expressed recombinantly, antigens from both pathogens can be expressed in a single attenuated capripox vector (as previously characterized by the Authors).  
  2. Please provide a detailed description of all controls employed in this study. Please describe control chemicals, control animals, control ‘vaccine’, etc. These must be included in a separate sub-section in M&M. The Authors have added the following to the M&M:

     “For infection experiments, the Authors used DMEM media as a negative control.  For viral genome quantification, viremia and viral load experiments, tissues from uninfected animals were used as the negative control.”

  1. How did you select and how did you include animals in the study? Please give a list of the criteria. Boer cross goats and Rideau Arcott sheep were selected, as these animals have been previously used to characterize the pathogencity of capripoxvirus, PPRV and RVFV at the Centre for Foreign Animal Diseases (Winnipeg, Canada).  In addition, all these diseases are not present in Canada so all the animals used in the studies are free from infection and demonstrated by serology. Therefore, based on our previous experience in optimizing factors such as: 1) the dose and route of infection; 2) knowing precisely when the onset of infection is supposed to happen, and; 3) the type of pathology anticipated with each infection, we felt that these optimized models would be ideal in evaluating the protective efficacy of our multivalent vaccine candidates. 
  2. Visualization for this manuscript is poor. You need to include more graphs with the results and also tables, whilst at the same time you must reduce the text in the manuscript. We would like to thank Reviewer #4 for pointing this out.  As also mentioned by Reviewer #2, graphs such as that presented in Figure 2 have been made larger in order to make it easier to read.  Furthermore, a schematic describing the vaccine candidates have also been added as a Figure. 
  3. Discussion. The Discussion is a bit shallow and does not fully cover all the findings of the study. Please expand and please describe in greater details the clinical benefits from using this vaccine. The Authors appreciate the criticism raised by Reviewer #4.  As the Authors have elaborated on several topics raised by the other 3 Reviewers, we believe the the Discussion section has been expanded (and articulated) significantly, in order to better explain the Discussion section.
  4. Do you think that the vaccine can be commercially available for use by the end of October? This will be very helpful for farmers…..While the Authors would love to have this vaccine available by the end of October, a larger-scale field study still needs to be performed. While the Authors are currently looking into performing such as study, it is very likely that it will not be completed by the end of this year.
  5. References are OK…..Thanks.
  6. Concluding section is missing. The Authors thank Reviewer #4 for pointing this out.  The Authors have now added a concluding section, following the Discussion section. 

Round 2

Reviewer 1 Report

Comments and Suggestions for Authors

1) Authors’ response did not address the problem. “LnRP and LtpaRP vaccinated animals developed neutralizing antibodies 2 days and 4 226 days earlier than control animals after PPRV…” This sentence is still there, and there is no explanation for why the authors think this is important. It appears not to correlate with anything – not Figure 2a or with the results for the negative control. There is also no indication that the scoring of animals was blinded to the scorer, so the comment about malaise is not relevant unless we know the scoring was blinded.

2) Authors’ response again did not address the problem. You can’t have it both ways: if the results in Figure (now) 5 “demonstrate that genome copies does not correlate to viremia,” the hypothesis that the results in Figure 3a are a result of difficulty in detecting viremia rather than a lack of correlation must be demonstrated in some way. In the absence of data, we can only assume that there is no correlation and no infective particles. The answer “the authors hope that the genomic data can still provide strong evidence for protection in immunized animals” then contradicts the conclusion in Figure 5.

3) If the results in Figure 4 show no statistical significance, then state it in the text.

All other Reviewer comments have been addressed sufficiently.

Comments on the Quality of English Language

Standard copy-editing should be sufficient to correct any English errors.

Author Response

Reviewer #1

1) Authors’ response did not address the problem. “LnRP and LtpaRP vaccinated animals developed neutralizing antibodies 2 days and 4 226 days earlier than control animals after PPRV…” This sentence is still there, and there is no explanation for why the authors think this is important. It appears not to correlate with anything – not Figure 2a or with the results for the negative control. There is also no indication that the scoring of animals was blinded to the scorer, so the comment about malaise is not relevant unless we know the scoring was blinded.

The Authors would like to point out that the attending veterinarians and technicians (i.e. the ones who did the scoring) did not know which groups were which—only which animals were associated with each group number.  Therefore, the scoring was performed blind.  In order to address this concern, the Authors have added the following to the Materials and Methods section (lines 155-158):

“The scoring of clinical signs were performed by the attending technicians and veterinarians with no prior knowledge of the which animals belonged to which experimental group, thereby conducting their observations blind to each vaccination group”. 

With Regards to Figure 2a, the Authors have simply decided to modified the sentence in question (lines 219-223) While vaccination alone with any of the three vaccines did not induce detectable levels of PPRV neutralizing antibody (Figure 2b), all animals developed neutralizing antibodies following PPRV challenge.

2) Authors’ response again did not address the problem. You can’t have it both ways: if the results in Figure (now) 5 “demonstrate that genome copies does not correlate to viremia,” the hypothesis that the results in Figure 3a are a result of difficulty in detecting viremia rather than a lack of correlation must be demonstrated in some way. In the absence of data, we can only assume that there is no correlation and no infective particles. The answer “the authors hope that the genomic data can still provide strong evidence for protection in immunized animals” then contradicts the conclusion in Figure 5.

The Authors apologize if they did not make their point clear in the first round of revision.  As the authors tried to state previously, the PPRV infection model that they established in a previous publication (Truong, T., Boshra, H., Embury-Hyatt, C., Nfon, C., Gerdts, V., Tikoo, S., Babiuk, L. A., Kara, P., Chetty, T., Mather, A., Wallace, D. B., & Babiuk, S. (2014). Peste des petits ruminants virus tissue tropism and pathogenesis in sheep and goats following experimental infection. PloS one, 9(1), e87145.) was used to evaluate the efficacy of these vaccine candidates.  In this publication, the Authors could not detect any infectious particles oral and nasal secretions from PPRV-infected animals.

Meanwhile, our group (as well as other groups around the world) were able to isolate infectious RVFV particles from experimentally-infected animals.

Therefore, with regards to “trying to have it both ways”, the authors are doing anything but that—they simply are using techniques previously validated for PPRV and RVFV experimental infections  (which are different).  Considering that both viruses come from different families, are transmitted differently, show different tropism and pathology, one can’t be too surprised if their detection methods differ in experimental models.  As previously stated, infectious particles for PPRV could not be detected in the nasal and oral swabs of experimentally infected animals, so performing an experiment trying to do so would not yield anything.  

3) If the results in Figure 4 show no statistical significance, then state it in the text.

The Authors have added the following sentence (lines 283-285):

Despite the qualitative data represented in Figure 4, it should be noted that no statistical significance was observed, likely due to the small sample size in each experimental group.

All other Reviewer comments have been addressed sufficiently.

Thank you for your comments.

Reviewer 4 Report

Comments and Suggestions for Authors

The authors have addressed all the points raised in previous evaluation. No further comments.

Author Response

The authors have addressed all the points raised in previous evaluation. No further comments.

Thank you for your response.

Round 3

Reviewer 1 Report

Comments and Suggestions for Authors

We will agree to disagree. Good luck with your vaccine studies.

Comments on the Quality of English Language

Some proofreading is needed.

Author Response

Comments and Suggestions for Authors

We will agree to disagree. Good luck with your vaccine studies.

Response: Thank you for reviewer the paper and your insights.   Comments on the Quality of English Language

Some proofreading is needed.

Response: We have proofread the paper one last time.